# The Experiences of African American Male Caregivers

**DOI:** 10.3390/healthcare10020252

**Published:** 2022-01-28

**Authors:** Barbara Pollard Deskins, Susan Letvak, Laurie Kennedy-Malone, Pamela Johnson Rowsey, Leandra Bedini, Denise Rhew

**Affiliations:** 1Cone Health, Greensboro, NC 27401, USA; Barbara.Deskins@conehealth.com (B.P.D.); Denise.Rhew@conehealth.com (D.R.); 2School of Nursing, UNC Greensboro, Greensboro, NC 27402, USA; lmkenne2@uncg.edu (L.K.-M.); pjrowsey@uncg.edu (P.J.R.); 3School of Health and Human Sciences, UNC Greensboro, Greensboro, NC 27402, USA; labedini@uncg.edu

**Keywords:** caregiving, African American males, COVID-19 pandemic

## Abstract

Aging of the population has led to an increasing number of caregivers. While research has been conducted on caregiver experiences, less is known about the experiences of African American males in the U.S. This qualitative descriptive study describes the experiences of 13 African American men who acted as caregivers to adult chronically ill or debilitated loved ones, the majority of whom provided care during the COVID-19 pandemic. The revised Sociocultural Stress and Coping Model (R-SSCM) guided this study. Individual interviews were conducted via the Zoom application. Content analysis revealed four themes: (1) “My time to take the reins”, (2) “It’s a male thang”, (3) “Caring amid a pandemic”, and (4) “Effects of the caregiver’s journey”. This study’s findings in relation to the literature provide overdue attention to the African American male caregiving experience, especially in relation to the COVID-19 pandemic. The implications of our findings include the need for more culturally congruent support services for African American male caregivers, as well as increased efforts to encourage trust in the healthcare system—especially in relation to vaccination for the SARS-CoV-2 virus.

## 1. Introduction

The United Nations (UN) reports that for the first time in human history there are more older adults than young; in 2019 there were 703 million adults over the age of 65 worldwide, which is expected to double to 1.5 billion by 2050 [1]. Living longer also increases the likelihood of needing caregiver assistance in older age. The American Association of Retired Persons (AARP)’s 2020 Report on Caregiving [2] documents that in the United States (U.S.) an estimated 48 million Americans are caregivers to an adult. While women have historically been the primary caregivers of older family members, more men are increasingly assuming roles as family caregivers [3]. In the U.S., 39% of caregivers are now men and, of this group, 34% identify as African American [2]. While the average age of all caregivers is 49.9 years of age, the average age for African American caregivers is 47.7 [2]. Additionally, African American caregivers are more likely to have lower incomes and be unmarried [2]. Although the non-refereed literature is abundant with findings regarding the increasing role of male caregivers in our aging society, the contributions of African American male caregivers have been underrepresented in peer-reviewed publications [4,5]. Gaps may exist due to the lack of African American male caregivers’ awareness of being family caregivers. Male caregivers in the Black et al. [6] study on elderly spousal caregiving responded “no” to the caregiver label’s inquiry; instead, they described themselves as “just a loving husband” or “devoted son” and “doing what I’m supposed to do”. Caregiving is known to have adverse effects on the caregiver’s health [7]. This is especially significant for African American men, because in the U.S. this group exhibits the highest mortality rate and the worst health profile compared to other racial/ethnic groups of men, due to structural and interpersonal racism [8]. Research on African American male caregiver experiences is substantially underrepresented in current empirical research [5]. Within the caregiving body of literature, there is a knowledge gap regarding the roles and experiences of African American male caregivers—especially during a pandemic. For these caregivers, the relative lack of discourse around self-care and wellbeing continues to evolve as the impact of COVID-19 takes its toll across racial/ethnic groups. Thus, the aim of this study was to explore and understand the experiences and perspectives of African American male caregivers, including during a global pandemic.

## 2. Materials and Methods

### 2.1. Study Design

This study utilized a qualitative descriptive approach. Qualitative research is applicable in studies when researchers aim to share individual stories of participants, use a literary or flexible writing style, and understand the context or setting of issues [9]. According to Sandelowski [10], the qualitative descriptive approach presents the phenomenon of interest “in everyday terms of those events (p. 336)”. These studies are interested in discovering the “who”, “what”, “where”, and “why” of events or experiences [11]. This study captured “rich data” through the experiences of African American male caregivers, and provides a more explicit understanding and meaning of their experiences from their perspectives, and with their narratives.

### 2.2. Participants

For the purposes of this study, a male caregiver was defined as a self-identified, male-gendered person who provides care (activities and experiences that involve help and assistance) to a dependent, chronically ill adult care recipient who is unable to provide care for themselves. The inclusion criteria for male caregiver participants in the study included (a) African American men over the age of 18 years caring for an adult family member with a chronic illness; (b) self-reporting as the primary caregiver for at least 6 months, or within 2 years of the caregiving experience; (c) English-speaking; (d) able to understand the purpose of the study and agree to participate voluntarily; and (e) willing to participate in an interview for 45–60 min or longer, via telephone or mobile phone, or a form of social media (e.g., Zoom). Exclusion criteria were if the male caregiver functioned in a secondary caregiver role.

After receiving IRB approval at the researchers’ university, African American male caregivers were recruited using purposive sampling. According to Bradshaw et al. [11], purposive sampling selection is based on accessibility to the participants and the researchers’ ability to select participants whose qualities or experiences can answer the research questions. Researching family caregivers of chronically ill family members poses many challenges, including finding caregivers with time to participate, despite their caregiving responsibilities [12,13]. In the present study, recruitment of research participants was directed towards the central part of North Carolina, and occurred via recruitment flyers posted in public areas, such as barbershops, adult daycare senior centers, African American churches, and a caregiver book promotion by WebEx (Milpitas, California, USA). The researchers also enlisted the assistance of one local adult daycare program social worker, who served as the liaison between the program’s male caregivers and the researchers. The social worker was instrumental in sending all study materials via email to a select group of caregivers who they believed would be willing to participate in a research study. Social media was also used to recruit participants, which led to participants outside of central North Carolina, but still within the southeast of the U.S. An advantage to using social media for recruitment is the researchers’ ability to reach hard-to-reach individuals [14]. The social media platforms Facebook and LinkedIn were used from January to April 2021 to recruit the men who self-reported as primary caregivers to a dependent family member.

### 2.3. Data Collection

Data were collected over eight weeks in February through April in 2021. Due to the COVID-19 pandemic, interviews were conducted using the Zoom application. Zoom is a collaborative, cloud-based videoconferencing service offering features including online meetings, group messaging services, and secure recording of sessions [15]. Zoom provides the ability to communicate in real time with geographically dispersed individuals via computer, tablet, or mobile device. Zoom’s key advantage is its ability to record and store sessions securely, without recourse to third-party software. This feature is particularly important in research where the protection of highly sensitive data is required. Other critical security features include user-specific authentication, real-time encryption of meetings, and the ability to back-up recordings to online remote server networks (“the cloud”) or local drives, which can then be shared securely for collaboration [15]. Interview times and dates were selected by study participants in order to minimize possible disruptions of the caregivers’ schedules. The instrumentation used for the study consisted of a semi-structured interview guide (Table 1), a demographic questionnaire (caregivers and care recipient data), and visual facial observation of study participants. The primary researcher conducted all interviews, which lasted from 49 min to 99 min. Each caregiver was offered USD 25.00 for his participation. Data were collected until no new information was being heard (data saturation).

### 2.4. Data Analysis

The revised Sociocultural Stress and Coping Model (R-SSCM) guided data analysis for this study. The qualitative content analysis strategy of Elo and Kyngas [16] was used to analyze transcript data. Steps included a (a) preparation phase, (b) organizing phase, and (c) reporting phase. During the organizing phase, open coding, creating categories, and abstraction took place. Each identified meaning unit was labeled with a code that should be understood in terms of the context [16]. The categories provided a means of describing the phenomena, increasing understanding, and generating knowledge. Within this process, the researchers used an inductive approach to evaluate the data. When generating initial codes, the researchers used reflexive journaling, and developed a coding framework or codebook. Using keywords or phrases that were bolded or highlighted facilitated the creation of a visual and contextual interpretation. The development of themes was a process that was built from abstract codes to patterns, categories, and themes from the “bottom-up” [17]. This process meant working back and forth between the themes and the database until a comprehensive set of themes emerged, allowing the data analysis to remain “true” to participants’ accounts, and contributing to ensuring the transparency of the researchers’ interpretations [11]. Analyzing and synthesizing codes into themes was completed based on the research objectives. In the reporting phase, findings were reported using thick, descriptive language that included participants’ direct quotes.

Lincoln and Guba’s [18] trustworthiness criteria were utilized throughout the study’s research process. Readers can ascertain a researcher’s credibility through (a) prolonged engagement, (b) persistent observation, (c) triangulation, (d) peer debriefings, and (e) member checks. In the present study, the researchers demonstrated credibility through member checking and collaboration with research team members. To establish an environment that supported honesty among the caregivers, participants were informed of their right to discontinue participation or withdraw from the study at any point; no participants withdrew from the study. To ensure the dependability of the study’s results, accurate data of the telephone and Zoom web-based interview recordings were transcribed to narrative text and subsequently read and re-read in order to ensure accuracy before undergoing the data analysis process. Transferability refers to the generalizability of the inquiry. In this study, thick descriptions are provided by using direct quotes given by the study participants. Confirmability was achieved through the objectivity of the data’s accuracy and meaning, demonstrated by how the researchers provided detailed methodological descriptions, which included accurate records of contacts and transcription of interviews. This process allowed for an audit trail through accurate field notes, transcripts, and reflexive journaling.

## 3. Findings

Thirteen African American men, who were recent (*n* = 1) or present (*n* = 12) primary caregivers to an adult care recipient, participated in the study. All participants lived in the southeastern region of the U.S. Participants were recruited until data saturation occurred, defined as the point at which no new themes emerged [19]. Table 2 depicts the demographic characteristics and self-reported health ratings of the male caregivers participating in this study, as well as their care recipients.

Data analysis revealed four major themes: (a) My time to take the reins, (b) It’s a male thang, (c) Caring amid a pandemic, and (d) Effects of the caregiver’s journey; each of the four themes was subdivided into subthemes.

### 3.1. My Time to Take the Reins

The theme *“My time to take the reins”* emerged in response to the question regarding role assumptions. From the caregivers’ statements, three subthemes emerged for this theme: Do what I gotta do, The heart of caring, and Spirituality/Religiosity. 

### 3.2. Do What I Gotta Do

The experiences that make up this category were based on the caregivers’ roles and responsibilities; it also captured what decisions influenced the need or willingness to assume the role. One son described his reason for taking on the role:

*“You know, as we get older, you’re gonna be taking care of us, and you’re gonna have to take care of your brother after we pass on”.* Another participant who was a caregiver to his mother stated *”I came to the realization that there wasn’t anything that I wasn’t going to do. I don’t know why I can’t explain how I came to that. But I’ve always given my mother a bath. I’ve always clipped her toenails and sanded the bottom of her feet. I’ve always helped get her dress; you know. So, it’s in-depth, you have to do everything from cleaning to cooking to (to) translating to transporting. Uh, it’s 24 h a day”.*

### 3.3. The Heart of Caring

The next subtheme is based on the collective narratives that described the pulse of the study participants’ caregiving experiences. The experiences of these caregivers were exemplified through the emotions (verbal and nonverbal expressions) observed during the interviews. For example, one participant stated:

*“It’s my responsibility… it’s my responsible thing, for me to take care of my mom. She’s been taking care of people all her life, and I’m going to take care of her now”,* whereas another caregiver shared, *“To be a caregiver is truly an unselfish role. This is a role for the unselfish. It’s not just anyone can do this. … you have to be unselfish. And think of yourself less”. Additionally, this participant shared, “The benefits of it [caregiving]… is to never have to look back and say that I wish I could have could have done this or done that. And to being able to feel good about… It ain’t always the leaning tree that falls. But right now, she’s the leaning tree”.*

### 3.4. Spirituality/Religiosity

Most of the male caregivers described a connection to some higher power or “being”, whether as a coping strategy, or as a sense of commitment, spirituality/ religiosity, or faith, as exemplified in the analysis. Most of the men used their dependency on their religion or spiritual belief as a coping strategy, as well as being instrumental in taking on the commitment of being a caregiver to their family member. For example, one participant stated:

*“But you know, he [GOD] says, ‘honor my mother, my father, honor thy father, though mother, as I have commanded’. So, that was… that piece for me was not an option. It was a command from God.”* A different participant stated, *“I just feel as though the (the) reward comes from Lord letting him know that I did all I could do to the best of my ability”. He further shared, “God will help me through it. And that has kept me … those kinds of things keep me centered, just knowing where to go… and a no-quit attitude.… and I said, without God in my life, I would have made a different decision. But because of that commitment to God, and her at the same time, it made all the difference. And… those seeds… see, this is the reality when I… when I give testimony… that’s… I tell people, the God that I serve is the reason I do what I do”.*

### 3.5. It’s a Male Thang

This theme was represented by four subthemes: coping, cultural aspects, societal perspective, and support systems from the male caregivers’ point of view. All caregivers provided some response as to how they coped with the stress of being caregivers. 

### 3.6. Coping

Within this category, eight strategies were identified that described how the caregivers reacted to or managed their stress levels. The strategies described were (a) faith/spirituality/religion; (b) relaxation techniques; (c) exercise; (d) non-stress-related paid work; (e) “me time;” (f) respite; (g) professional/informal emotional support; and (h) anger management. Coping statements included:

*“I’ll just put it in a nutshell, finding rest, finding peace, finding comfort, the ability to cope, I found it through messages and reading the Bible, and things like that would basically help bring me down to where I needed to be”.* Exercise as a means of coping included taking walks, attending a gym, yoga, or bowling. As told by one caregiver: *“I get outside of myself, you know, uh… That’s what I do… is I busy myself, I move a muscle, change a thought, to the best of my ability”.* Three of the caregivers provided direct statements that underscored the “me time”, described as “alone time”, “peace and quiet”, “fishing”, “you time”, “meditation”, and “long baths”.

The respite coping strategies captured the time the caregivers attended to self-care needs or relaxed. This opportunity was provided when the care recipients participated in the adult day care center, paid out of pocket for sitters, or members of the family or friends relieved the caregiver. Eight of the participants described some form of *“away time”.* Professional or informal emotional support was another form of coping strategy. Several caregivers discussed how they sought professional help or leaned on others who were held in high regard. For example, one participant stated:

*“So, my coping mechanism was to listen to that (that) therapist who said (caregiver’s name), you need to plan some time by yourself … for yourself… with things you enjoy”.* Another stated *“I talk with two of my friends. One is a psychiatrist, out of (university name). I have another one who is a therapist”. In contrast, one caregiver opposed the need for professional assistance. He gave this response: “No professional, not (not) okay? I won’t go unless I’m feeling a little something, something now. That’s the only reason I will go. And it’ll be for everybody else’s safety… not mine (laughter)”.*

### 3.7. Cultural Aspects

Culture encompasses the social behaviors and norms of a group of people; it also embraces the knowledge, beliefs, and customs of an ethnic group. For some of these African American male caregivers, culture tended to define the reasoning for their sacrifices and their determination for assuming the caregiver role. For example, a participant offered these remarks:

*“I was brought up, you know, to take care of your family. And I did come up with, you know, the man being the leader, the head of the house”.* Meanwhile, another caregiver believed it was a son’s duty or a child’s duty to look after their parents: *“So, it’s a dutiful; it’s a dutiful thing for me as well”.* Lastly, one son offered the following statement: *“There needs to be some immediate awareness made that it’s okay to decide to do your best to take care of your loved one… from a male perspective. It doesn’t make you weak; it doesn’t make you less of a man”.*

For many participants, stigmatization has been associated with Black and racial discrimination or the stigma of women’s work. As a survival strategy, the African American population tends to remain unified as a culture or within their ethnic group. This is best exemplified by one participant, who stated:


*“And I think, you know, the cultural aspect of being Black in America. And being a Black man in America, and the stereotypes around that, you know. How do I… like I said, I happen to be very lucky. Um, but I really don’t talk to anybody. In our culture, it is a woman that does the work [of caregiving], it’s, you know, you’re gonna, you’re… most of friends of mine, their wives take care of their mothers. I think that’s got to change; we’ve got to open up, we’ve got to evolve in our culture”.*


### 3.8. Societal Perspective

Historically, caregiving’s connotation as “women’s work” has reflected society’s view of who have been the primary informal caregivers within families. Despite the increasing emergence of male caregivers, society has been slow in accepting this trend, and maintains traditional caregiver norms.


*“Most the time, women… you expect for women to be the caregiver. I think that’s what most folks expect”, as uttered by one participant. Another voiced “in this… today’s society, we have created a society where we are giving women a role to play and given men a role to play”; he also added “If you read the studies on girls and boys, boys are the most sensitive, which is why I think we’re taught to ‘man-up’ at a certain age, it’s the young boy that will go and hug their mom and go and hug their dad a little bit more than the young girl would, you know, do as children. So, men are compassionate… we’re just taught not to be”.*


### 3.9. Support Systems

Supportive services and networks were notable resources identified by the 13 male participants. Reluctance to seek formal assistance was noted by two of the participants. For example, one participant stated:


*“I’ve had ‘em… I’ve had them trying to tell me how to run the house. Tell me what to get. Tell me what to do. And I’m like, wait a minute. Excuse me; you got it wrong. So, me personally, if I had someone to come in and take care of my mom. I wouldn’t want it because of what I’ve been through with caregivers. That’s just being very honest”. To add to this participant’s reflections of informal support, he further stated “Church… umph… you know, the church really bothers me. Because they always talk about calling, and gonna come see, and doing this and do that. And to be honest, they don’t do anything”.*


However, the majority of the participants described use of formal and informal supportive networks. Examples of formal care services included home health, hospice, nutritional support, and adult day care services. Informal support networks included spouses, siblings, caregiver dependents/children, and support from friends and church members. The following statement describes one participant’s informal support:


*“My daughter is a big help. If I call my daughter and say I need you to take (take) your mom for a little while, she usually does… no matter what she has to do. Grandchildren, the kids are old enough to be able to help her do those things”. Another stated, “So, my brother can come see my mother, cheer her up, he’s happy, go-lucky… come and go. My sister… she was involved with her kids heavily. So, we would make plans to where she would go there for the holiday or (or) stay for a month or two to give me a break or things of that nature. So gradually, my sister participated”.*


Several of the caregivers commented on personal benefits from support groups; however, they did not attend a support group. One participant stated: 


*“I’ve been debating it, but I just haven’t had time to do that right now. It probably would be good for me because I could hear from other people. But I don’t know of any, and I haven’t really tried to pay attention to it. I think there’s some Alzheimer’s groups around someplace”. Another stated “I never thought about it. Not too many men, and I know, do caregiving… or even think about it”. Finally, “I don’t think we have in our mind that there are support services available to us… And then the perception may be that if I avail myself to those services, how will I be looked at by my family and other people?”*


### 3.10. Caring Amid a Pandemic

Twelve of the thirteen caregivers were caregivers during the COVID-19 pandemic, and shared their caregiving journeys. The one caregiver who did not provide care during the pandemic was still included in this study, since he met all inclusion criteria. The questions asked of these participants included the following: (a) if and to what extent were he and the care recipient affected, (b) the willingness to accept the vaccine for himself and/or for his loved one, and (c) pandemic-specific effects on his social interactions with others. It should be noted that the willingness to take the FDA-approved COVID-19 vaccine became a point of inquiry, since the vaccines were available. Significant responses included: (a) protection from the virus and spreading to their loved one, (b) changes in formal and informal support systems, and (c) lifestyle changes. Several responses are described, as follows:


*“When I wash dishes, for example, I normally use hot, hot (hot) water and everything. But now, COVID, I also put some Clorox in the water because I want to make sure that that even if I’m washing dishes thoroughly, that they’re thoroughly sanitized. We do (do) things to try to try to head off this COVID if it should happen to get around us”. Another participant stated “She (aide) comes with her mask; she puts a mask on; granny was putting hers on too. When stuff coming in, but I’m just saying my keys, my wallet, the doorknobs? I wipe the car down and all that I’m trying to do as best we can, you know, we keep wipes around in the house”.*


One participant recounted a hospital encounter when he was not allowed to stay with his disabled spouse (care recipient) due to the COVID-19 restrictions. He responded:


*“So, now I go to the hospital. And they tell me, I can’t, I can’t even stay with my wife. They tell me. And I gotta adjust to that?”*


Several of the caregivers shared changes in their formal and informal support systems due to the pandemic restrictions. For instance:

*“That’s why I said,… this summer… this summer was rough with my mom. It really was… uh… no (adult center). The only company she was getting was her neighbors. And it was, it was (it was) something”.* Another participant stated *“They ask, can they come and do things but because of COVID, I don’t allow anyone in the house … because of my mother’s numbers” (care recipient has leukemia). Adding to this response, he stated “I’m just afraid that, uh, somebody’s coming in … they still have to go home to their lives. So, I don’t allow anyone (with emphasis) in my home”.*

During the data collection timeframe for this study, the COVID-19 vaccine was in the immediate phases of availability to the highest risk category of individuals, which included older adults and caregivers. Therefore, questions were asked regarding accepting the vaccine. Feedback regarding this line of inquiry is summarized as follows: two caregivers said they would accept the vaccine later (to see how others were affected); two other caregivers responded that they would not take the vaccine; several others responded that they would take the vaccine without question.


*“I trust that when they say it’s ready (COVID vaccine), I’m ready. And so, I will be one that would take it… and Granny and (wife’s name) whenever they do it”. When faced with adverse comments from others regarding the vaccine, this participant stated “We all have to be at peace with whatever decision we make.… I said that’s fine… I say… I’ll be in line when it’s time to take the shot (laughter)”.*


Hesitancy to consider the benefits of accepting the vaccine was also identified in this group of study participants. One participant stated:


*“Yeah, yeah… we’ve (we’ve) been … our community has been experimented on so many times. And I’ll just talk about the Tuskegee is there’s other things in history as shown that, that that that that that we’ve been? We’ve been targeted at times”. Another stated “So, I wouldn’t consider a vaccine that was created in a couple of months; that I feel is only geared towards people of color and poor White folks. I think we’re being used as test objects with that”.*


Participants shared how the COVID-19 pandemic was impacting their social interactions. One caregiver stated


*“I know people invite me to cookouts and stuff like that, right? With this COVID thing going on. I say, ‘not my thing’. I don’t, (I don’t) (I don’t) get around a lot of folks. Plus, I can’t afford to bring anything back here to her. It makes you close your (close your) ‘circle the wagons’. It makes you close your (your) interaction with people. It’s closed to a certain (certain) group, okay? And then it closed my… closed mine even a little more. Because I can’t (I can’t) even … I can’t (I can’t) afford to bring some… or let someone bring something by here that does she don’t need … doesn’t need to get”. Another participant shared “Immediate family still comes over; otherwise, no other interactions. They come now (church members), they just come from the cars and leave; drops something off. So, there’s… everybody is wearing masks”.*


### 3.11. Effects of the Caregivers’ Journeys

This final theme includes three subthemes: (a) “stressors—positive and negative effects”, (b) “physical/emotional adverse health outcomes”, and (c) “benefits/gains—positive health outcomes”. 

Following the caregiver literature, caregiver stressors accounted for the most significant physical and emotional stress for these caregivers. Specific factors include co-residing with the care recipient, social isolation, financial difficulties, increased duration and length of time spent caregiving, absence of coping skills, problem-solving challenges, and lack of choice in being a caregiver [20]. Psychological adverse effect symptoms were described by the caregivers as including guilt, burden, feelings of abandonment, isolation, and frustration. Statements that further illustrate these effects are described as follows:


*“Sometimes I just, uh… (umph)… don’t know, no, kind of rages… so, I’m not gonna say that I feel bad. But a lot of times, I just, you know, feel some… sometimes defeated, as to not being able to uh… you, know… stop this from happening (progression of mother’s dementia)”. Another participant stated “I might get stressed out. Might get a little frustrated. So, me as a caregiver, we sometimes take a lot of that verbal abuse because of their inability to do what they used to do. Sometimes when you take care of your loved ones, they seem to… sometimes can remember… sometimes the bad and not the good. What you have to be a caregiver, you have to just kind of keep pushing on… strive past it and kind of in one ear out the other”.*


Physiological adverse effects were noted, including reported lack of sleep, fatigue, tiredness, exhaustion, weight gain, and anger. The caregivers described impaired health behaviors or lack of self-efficacy, such as neglecting one’s own healthcare needs. Emotional or mental health effects (e.g., depression) are extensively cited in the literature. Conversely, none of the caregivers used the words “depression” or “feeling depressed” outright in their responses. Examples of caregiver statements that detailed more of their self-reported physical adverse effects are described as follows: *“I never realized that I had anything to do with sleep apnea. I thought snoring was good sleep. I didn’t know it’s a respiratory problem”*. Another participant shared these comments regarding the physical effects on his body as a result of being a caregiver:


*“And now… trying to get back into things, I got it, ‘cause stressors caused some cosmetic changes in me. So, I have to get some things done to me visually. And just the level of stress being a diabetic also, losing teeth because of the clenching and my mouth and the grinding. I am recovering from partial sight… eyesight from stress (points to the eye)”.*


Caregiving can be associated with personal benefits to caregivers’ health outcomes. In this study, the researchers regarded health outcome benefits or gains, during the caregiving timeframe, as self-reported physical or emotional improvements. The researchers saw these as optimization of coping strategies or stress-reduction techniques. Five of the participants reported direct or indirect improvements in their health status. For example, one caregiver had stabilization of sarcoidosis after he took his mother out of a nursing home to provide care in her home. He stated:

*“I was commuting back and forth to (a city named). But because of the stress of that nursing home, commuting to (a city named) every day, except when I’d visit night shift, and then having to come straight from (a city named) and go help take care of my mother (which exacerbated my sarcoidosis)”.* Another caregiver found a way to reduce his stress by learning to take time for himself; *he stated “Sometimes I would plan days that I would leave the house, there would be a movie that I wanted to watch. I like watching movies and just sitting eating popcorn. And I would do those days, no one knew. Those days (were for) me. And I would just do them. That was, that was, my time”. Improvement in health is best exemplified by the caregiver who stated “We take care of each other really… It’s not that I just care for her because she helps me too emotionally. She’s a mother”.*

In summary, data analysis revealed four major themes. Table 3 provides a summary of significant themes and subthemes which informed the theme development.

## 4. Discussion

This study adds to the body of literature on male caregiving, and especially gives voice to the experience of African American male caregivers. First, participants in this study discussed their reasons for taking on the caregiving role. The present study’s findings showed that taking on the caregiver role was not an unexpected decision for any of the study participants except one. The chronic, deteriorating health of parents or spouses occurred over time before dependency on others was necessary. Although the men had time to prepare for the assumption of their role, none of them seemed prepared mentally and/or physically for the challenges of caregiver duties and responsibilities. An important finding in this study pertained to the “caregiver title” or “label” as perceived by the study participants. Given the current caregiving crisis, male caregiving is expected to increase [3,5]. While over half (60%) of the men in this study were accepting of the title, the rest of the study participants were averse to the term’s use. For those caregivers who did not ascribe to or identify with the title, they offered societal perspectives on the term’s use: “A means to categorize”; “A bucket to put things in”; “Commercialized”; and “A professional term”. In previous research, the caregiver label for some male caregivers, as identified in the current study, was seen as a stoic approach to caregiving, leading to their resistance to identifying as caregivers [21,22]. Moreover, the label of “caregiver” made the act of “caregiving” seem obligatory. This finding was substantiated in the current study, as study participants voiced these same reservations.

As Robinson et al. [23] found, when it came to duties and responsibilities, male caregivers tend to struggle with the feminine nature of tasks such as cooking, cleaning, and personal care tasks such as bathing or changing an adult brief. However, contrary to previous studies [2,24], the present findings challenge the traditional assumption that men are not very involved in household work or personal care. In the present study, many male caregivers described discomfort with this aspect of personal care provision, but they completed the tasks. Only two of the caregivers responded that they would not engage in personal care for their mother but managed other roles.

Prior research would suggest that many of this study’s participants displayed a gendered approach to caregiving, as seen in the form of hegemonic masculinity or stoicism-type behaviors [22,25]. Spendelow et al. [25] postulate that this form of masculinity commences when males are placed in roles, such as caregiving, where nontraditional masculine norms require incorporation. Several of this study’s caregivers also voiced their views of the caregiving tasks as “woman’s work”, similar to previous caregiving findings [6,23]. However, contrary to these studies, these male caregivers argued against these old-fashioned ideas of masculinity associated with caregiving, and offered “it’s okay to show emotion and let loose without judgment” and “men should move beyond holding their feeling”.

The men in this study all verbalized effective coping styles. According to the American Association of Retired Persons/National Alliance for Caregiving (AARP/NAC)^2^, African Americans may cope with caregiving better because their culture enables them to feel more positively about caregiving. This stems from the deep sense of commitment and spirituality, wherein no new themes emerged [26,27]. In the current study, several men discussed their ability to reach out to family and friends and have a supportive network (e.g., co-workers, church members). Our findings also indicate that several men accepted their responsibilities as caregivers—especially those without siblings or family support—and adapted to the role. These findings reflect the cultural aspects of keeping the care recipient within the home (i.e., free of nursing home placement). This was reflected in the number of men who cohabitated or adapted their lives to manage their loved one’s care. Overall, managing stress was not a problematic finding in the study, based on the caregivers’ narrative responses or self-reported health outcomes.

Most of the study participants had limited knowledge of the various supportive resources available to them, such as national organizations and societies of specific diseases that offer multiple programs to ease the emotional burden on caregivers. One specific question asked of participants was about their awareness of respite care services. Caregiver respite can be one means for family caregivers to take time for themselves and possibly avoid the need to relinquish their caregiving role [24,28]. Plans for unpredictable emergencies or caregiver relief are needed in order to support these caregivers’ physical and emotional strains.

For most men in this study, their roles as caregivers were established for a substantial amount of time before the COVID-19 pandemic. Only one of the caregivers was not involved as a caregiver during the pandemic. According to recent research [29,30], the pandemic’s impact on people’s social interactions (social life or social isolation) was regarded as a significant stressor that led to mental health concerns or problematic health behaviors. In contrast, in the present study, the participants conveyed minimal disruption to their lifestyles, citing decreased or absent social activities before the pandemic.

A more concerning finding related to the COVID-19 pandemic was seen as the hesitancy of a few caregivers to accept the FDA-approved vaccine. This is especially alarming given the increasing death toll in communities of color due to complications from the virus [31]. While most caregivers were inclined to receive the vaccine immediately or later (“waiting to see how others fared”), nearly 40% opposed accepting the immunization. As SARS-CoV-2 continues to mutate into novel variants, it is imperative that caregivers and vulnerable care recipients—especially those who are African American—get vaccinated.

The participants in this study all described physiological and psychological adverse effects. This study’s participants refrained from using the term “depression” or its derivatives, and only a few described improvements in their general health since beginning the caregiver role. However, many of the caregivers reported symptoms of depression (i.e., inability to sleep, exhaustion, tiredness, tension). The researchers ensured that all caregivers were established with a healthcare provider, and the number to a 24-h crisis hotline was provided. These findings were similar to those of prior studies [4,32] that described caregiver symptoms as outcomes of depression and anxiety.

There is a need to further explore how African American male caregivers address the financial burdens associated with caregiving. Our sample represented demographic findings similar to other studies regarding education, health status, and employment. However, financial challenges existed due to annual income and caregiver demands. Over 38% of participants reported an annual household income of less than USD 50,000, and 15% reported USD 25,000 or less. Notably, none of the participants verbalized a change in employment status due to the pandemic. It is extensively documented that lower income and financial strain is linked to adverse health outcomes [24]. Thus, adverse health outcomes may be particularly problematic for African American male caregivers, who tend to have lower income levels.

Finally, this study has implications for further policy development in the U.S. There have been policy actions around the issue of caregiving in recent years. For example, the Credit for Caring Act of 2019 was enacted to provide working family caregivers with a non-refundable tax credit of up to USD 3000 to assist with out-of-pocket expenses related to caregiving. Caregivers can use this tax credit toward costs such as transportation, home modifications to accommodate a family member, medication management services, and training or education. However, additional policy efforts are needed in order to lessen caregiver financial strain, as well as to provide resources to support caregivers and vulnerable care recipients.

## 5. Limitations

There are limitations to this study. The sample was limited to African American men who met the study criteria geographically, located along the eastern region of the U.S. It is possible that expanding this study to other areas of the country would yield different results. Another limitation of this study is the effects of the COVID-19 pandemic on caregivers. The pandemic continued to evolve during the interviews, and is incessant at the time of this writing. Therefore, given the pandemic’s evolving nature, this study’s findings may not be generalizable to future studies that occur during a different time, or in a non-pandemic public crisis.

## 6. Conclusions

The demand placed on caregivers will continue to rise with the aging of the global and U.S. populations. The individual and collective stories of these African American male caregivers—an invisible caregiver group in the literature—can enrich other caregivers’ lives, male research studies, and interprofessional caregiver research. The personal stories of the 13 African American male caregivers participating in this study provide overdue attention and recognition to African American men in the healthcare research and caregiving literature. 

## Figures and Tables

**Table 1 healthcare-10-00252-t001:** Semi-structured interview guide for AA male caregivers of adult care recipients.

Concept/Topic Area	Guiding Questions
Caregiving Experience	What would you say it means to you, to be a male caregiver? What is it like for you to care for someone with a chronic illness or disability on a day-to-day basis?
Caregiver Role	Some people, even though they provide a lot of care to another person, do not consider themselves as a “caregiver”; why do you think that is? Do you think of yourself as a caregiver? Do others (i.e., family members/significant others) think of you as a caregiver? Please explain. Can you share something about how you came to take on the caregiver role for your loved one? When did you first start being the caregiver for your loved one? What sort of care do you help with?
Effects of Caregiving	Please share any difficulties you may have experienced since you began caring for your family member. Please describe what makes you feel stressed in your caregiver role. If you have stress, please describe how you cope with being a caregiver.
Support Systems	Tell me about the kind of support you get from your family or significant other Tell me about any support you get within the community (i.e., church, support groups, etc.). There are many services available to support caregivers in the community; what types of programs have you been made aware of? Have you accessed these services/programs? If no, can you share why not? What type of support services would you say you need to maintain your caregiver capabilities?
COVID-19	Tell me how being a caregiver during COVID-19 has (a) affected you personally (i.e., socially, physically, emotionally, or financially); (b) affected your ability to provide the same level of care to your loved one?
Summary	Thank you for taking the time to tell me about your caregiving experiences. Is there anything else you think I should know?

**Table 2 healthcare-10-00252-t002:** Caregiver and care recipient characteristics.

	N (%)	Min	Max	Mean	SD
Caregiver Age	13	38	77	58.9	9.13
Marital Status					
Married	6 (46)				
Unmarried	7 (54)				
Education Level					
High School	2 (15)				
Some College	4 (31)				
Bachelor’s Degree	6 (46)				
Graduate Degree	1 (8)				
Employment Status					
Full-Time	4 (31)				
Part-Time	3 (23)				
Retired	3 (23)				
Unemployed	3 (23)				
Income Level					
<USD 25,000	2 (15)				
<USD 50,000	3 (23)				
>USD 50,000	6 (46)				
Declined to Respond	2 (15)				
Caregiver General Health					
Excellent	0				
Good	7 (54)				
Fair	6 (46)				
Poor	0				
Years of Caregiving	13	1	24	7.46	6.83
* Care Recipient Age	16	58	100	81	11.91
Female	12	59	100	82.1	12.14
Male	4	58	89	78.4	12.30
* Care Recipient Disability					
Dementia	8 (61.5)				
Impaired Mobility	7 (64)				
Coronary Artery Disease	4 (31)				
Stroke	3 (23)				
Cancer	3 (23)				
Visual/Hearing Loss	3 (23)				
Brain Aneurysm	2 (15)				
Diabetes	2 (15)				
* Care Recipient Relationship					
Spouse	2 (13)				
Mother	9 (56)				
Father	3 (19)				
Sibling	1 (8)				
Grandparent	1 (8)				

***** Note: multiple care recipients for some participants.

**Table 3 healthcare-10-00252-t003:** Major themes and caregiver quotes.

Theme with Subthemes	Informative Quotes
1. My Time to Take the Reigns	
a. Do what I Gotta Do	
	“You know, as we get older, you’re gonna be taking care of us, and you’re gonna have to take care of your brother after we pass on”.
b. The Heart of Caring	
	“It’s my responsibility … it’s my responsible thing, for me to take care of my mom. She’s been taking care of people all her life, and I’m going to take care of her now”.
c. Spirituality/Religiosity	
	“But you know, he [GOD] says ‘honor my mother, my father, honor thy father, though mother, as I have commanded’. So, that was … that piece for me was not an option. It was a command from God.”
2. It’s a Male Thang	
a. Coping	“So, my coping mechanism was to listen to that (that) therapist who said (caregiver’s name), you need to plan some time by yourself … for yourself … with things you enjoy”.
b. Cultural Aspects	“Um, but I really don’t talk to anybody. In our culture, it is a woman that does the work [of caregiving], it’s, you know, you’re gonna, you’re … most of friends of mine, their wives take care of their mothers. I think that’s got to change; we’ve got to open up, we’ve got to evolve in our culture”.
c. Societal Perspectives	“In this … today’s society, we have created a society where we are giving women a role to play and given men a role to play”.
d. Support Systems	“My daughter is a big help. If I call my daughter and say I need you to take (take) your mom for a little while, she usually does”.
3. Caring Amid a Pandemic	“She (aide) comes with her mask; she puts a mask on; granny was putting hers on too. When stuff coming in, but I’m just saying my keys, my wallet, the doorknobs? I wipe the car down and all that I’m trying to do as best we can, you know, we keep wipes around in the house”.
	“So, now I go to the hospital. And they tell me, I can’t, I can’t even stay with my wife. They tell me. And I gotta adjust to that”.
4.Effects of the Caregiver’s Journey	
a. Stressors—Positive/ Negative Effects	“I might get stressed out. Might get a little frustrated. So, me as a caregiver, we sometimes take a lot of that verbal abuse because of their inability to do what they used to do”.
b. Physical/Emotional Adverse Health Outcomes	“And just the level of stress being a diabetic also, losing teeth because of the clenching and my mouth and the grinding. I am recovering from partial sight … eyesight from stress (points to the eye)”.
c. Benefits/Gains— Positive Health Outcomes.”	“We take care of each other really… It’s not that I just care for her because she helps me too emotionally. She’s a mother”.

## Data Availability

Data are available upon request to the authors.

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
