# Peer review of "The Experiences of African American Male Caregivers"

_healthcare, 2022, doi:10.3390/healthcare10020252_

Round 1

Reviewer 1 Report

The purpose of the paper is very interesting and useful, however, in this presentation the value is reduced, making this article an opinion article. I suggest making improvements in methodology and results, with comparative analysis between groups (pre and post covid-19;  male versus female, or between countries), presenting results ojectively in tables or graphs. 

An exclusive descriptive analysis has low scientific interest, although the idea is original.

Author Response

Reviewer Comment: The purpose of the paper is very interesting and useful, however, in this presentation the value is reduced, making this article an opinion article. I suggest making improvements in methodology and results, with comparative analysis between groups (pre and post covid-19;  male versus female, or between countries), presenting results ojectively in tables or graphs. 

Authors' response: 
    Thank you for your comments- we unfortunately used the template provided and were too brief in our reporting.  We have enhanced the methodology and discussion.  We have added tables for the quotes to organize the manuscript better. We are unable to provide comparative analysis since “comparing groups” was not a purpose of the study -indeed, Lindsay (2019) conducted a scoping review of qualitative comparison studies and identified many limitations to qualitative comparative studies. We agree this is an intriguing idea and will strongly consider it in future studies. 

Reviewer Comment: An exclusive descriptive analysis has low scientific interest, although the idea is original.

Authors' response: We recognize that unfortunately qualitative research does not hold the prestige of large scale quantitative designs and clinical research trials, however, scientific rigor was used in this qualitative study. Qualitative research provides valuable insight into the day to day experiences of individuals which is often lost in large quantitative studies. 

Reviewer 2 Report

The paper deals with caregiving with a focus on male African American caregivers. The study is based on interviews with 13 caregivers. The methodological basis of the interviews is appropriate. The main deficit is the very low numbers of interviews.

The authors claim to fill the gap of analysis of male African American caregivers. But, the authors do not highlight what are the special features of this group and what are the differences to other ones, e.g. male white American caregivers. At least this aspect should be elaborated in more detail.

Author Response

Reviewer Comment: The paper deals with caregiving with a focus on male African American caregivers. The study is based on interviews with 13 caregivers. The methodological basis of the interviews is appropriate. The main deficit is the very low numbers of interviews.

Authors Response: Thank you for your comment- we added support (data saturation) for our inclusion of 13 participants. Additionally, Vasileiou and colleagues (2018) recent systematic review of 15 years of qualitative research published in BMC Medical Research Methodology journal found that 12-13 interviews is sufficient as a sample size for qualitative health research. 

Reviewer Comment: The authors claim to fill the gap of analysis of male African American caregivers. But, the authors do not highlight what are the special features of this group and what are the differences to other ones, e.g. male white American caregivers. At least this aspect should be elaborated in more detail.

Authors' response: Thank you for pointing this out- we have enhanced our background information. 

Reviewer 3 Report

  • Last sentence of the abstract: This sentence is indeed true and shows the impact of your work. However, it seems more appropriate to finish the Abstract with a specific conclusion based on your findings, in addition to clearly pointing out the implications.
  • Introduction: In general, authors should add more information describing the demographics and circumstances of caregivers, in particular the African American male caregivers. In order for the reader to understand the knowledge gap, they need to be walked through what is already known on the subject, as scarce as it might be. Currently, this section does not provide that.
  • Lines 29-30: First, it is not appropriate to cite a blog article in a scientific publication, especially when you are providing numerical data. In fact, the blog refers to the report from which it took these numbers, and it is appropriate to cite the original document from which these data are taken. Second, this sentence is not correct - the blog which is cited does not state that out of all male caregivers more than 15% are African American, it states that among African American caregivers more than 15% are men.
  • Lines 32-34: Please consider rephrasing this sentence for clarity. Namely, as it currently reads this sentence states that there is a gap in the literature regarding the need for African American male caregivers. Given the aim of the study, the research question and used methods - it seems that this sentence should be rephrased. Based on the article, it seems as if there is a knowledge gap regarding the role and experiences of African American male caregivers.
  • Section Materials and Methods and subsections Participants and Data Collection are duplicated in the manuscript.
  • Within the Material and Methods section, the first paragraph could belong to the subsection "Study Design".
  • Section Participants: This section in fact does not (almost at all) describe the participants. It describes how they were recruited and how data was collected. From which area were the men recruited? How long were recruiting efforts made? What's the specific period in 2021 during which data were collected? How did you define a caregiver?
  • Section Data Collection: Specify that data were obtained on care recipients too.
  • Line 110: Define a recent caregiver. How many were recent and how many present caregivers? Do findings differ between them or is it justified to observe them together? In particular since the research question and title of your article refer to caregiving during the pandemic and later on you state one participant was not a caregiver during the pandemic. This is why it is important to define caregivers in the methods and also define inclusion criteria. If you were assessing the experiences of African American men who are caregivers during the pandemic, then it is not clear how you included a participant who did not provide care during the pandemic. If you were not assessing this - then the title of the manuscript needs to change because currently it emphasizes pandemic.
  • Lines 111-112: Table 2 does not show self-reported health ratings, is this information accidentally omitted from the Table?
  • Section Findings: It seems that for the sake of coherency, the first paragraph should be divided in two paragraphs, the second one should start from "Data analysis revealed..." and this text should follow the Table 2 placement, not go ahead of it.
  • Table 2: Correct the title, participants are in fact caregivers. In addition, cells for care recipients in the first column need to be revised. Further on - methods section states that demographics were collected, were age and the years of caregiving  the only two demographics you collected? Was there no information collected on socio-demographic characteristics? Education, employment, marital status, income etc.? If so, this needs to be noted as a limitation of this study. Also, you stated that you collected self-reported health rating - show it in this table.
  • Discussion: Please check whether the style of citing references in the text is in line with Journal's instructions.
  • Lines 420-426: In this paragraph, consider underlining the possible impact on care recipients too, as a vulnerable population.
  • Lines 434-437: Were information collected on this? These need to be presented in the Findings section, possibly within Table 2.
  • Lines 445-447: Define the study criteria for inclusion of men within the Methods section.
  • Lines 448-449: How was the effect of pandemic assessed for the one caregiver who did not have a caregiving role during the pandemic, as you've stated in the manuscript?
  • English language of the manuscript: Please check the text for minor grammatical errors. E.g. in the last sentence of the Manuscript (in Conclusion). 

Author Response

Reviewer Comment: Last sentence of the abstract: This sentence is indeed true and shows the impact of your work. However, it seems more appropriate to finish the Abstract with a specific conclusion based on your findings, in addition to clearly pointing out the implications.

Authors' response: Thank you- we have changed the abstract to include implications of the study findings

Reviewer Comment:  Introduction: In general, authors should add more information describing the demographics and circumstances of caregivers, in particular the African American male caregivers. In order for the reader to understand the knowledge gap, they need to be walked through what is already known on the subject, as scarce as it might be. Currently, this section does not provide that.

Authors' Response: Thank you for pointing this out. We have provided more information with specific percentages. 

Reviewer Comment: Lines 29-30: First, it is not appropriate to cite a blog article in a scientific publication, especially when you are providing numerical data. In fact, the blog refers to the report from which it took these numbers, and it is appropriate to cite the original document from which these data are taken. Second, this sentence is not correct - the blog which is cited does not state that out of all male caregivers more than 15% are African American, it states that among African American caregivers more than 15% are men.

Authors' Response:  Thank you for this comment- the blog reported on an AARP report.  We have gone to the original research report and have updated appropriately. 

Reviewer Comment: Lines 32-34. Please consider rephrasing this sentence for clarity. Namely, as it currently reads this sentence states that there is a gap in the literature regarding the need for African American male caregivers. Given the aim of the study, the research question and used methods - it seems that this sentence should be rephrased. Based on the article, it seems as if there is a knowledge gap regarding the role and experiences of African American male caregivers.

Authors' Response: Thank you for pointing out our lack of clarity. We have reworded for clarity.

Reviewer Comment: Section Materials and Methods and subsections Participants and Data Collection are duplicated in the manuscript.

Authors' Response: Thank you for picking this up- we apologize for the duplication and have removed the text. 

Reviewer Comment: Within the Material and Methods section, the first paragraph could belong to the subsection "Study Design".

Authors Response: Thank you- we have added a sub-section of Study Design.

Reviewer Comments: Section Participants: This section in fact does not (almost at all) describe the participants. It describes how they were recruited and how data was collected. From which area were the men recruited? How long were recruiting efforts made? What's the specific period in 2021 during which data were collected? How did you define a caregiver?

Authors' Response: Thank you for pointing this out- We added more information to the demographics table. We included information on the recruitment period in 2021 and inclusion criteria. We provided our definition of male caregiver. 

Reviewer Comment: Section Data Collection: Specify that data were obtained on care recipients too.

Authors' Response:  Thank you- we have updated that data were collected on care recipients as well and added their information to the table.

Reviewer Comment: Line 110: Define a recent caregiver. How many were recent and how many present caregivers? Do findings differ between them or is it justified to observe them together? In particular since the research question and title of your article refer to caregiving during the pandemic and later on you state one participant was not a caregiver during the pandemic. This is why it is important to define caregivers in the methods and also define inclusion criteria. If you were assessing the experiences of African American men who are caregivers during the pandemic, then it is not clear how you included a participant who did not provide care during the pandemic. If you were not assessing this - then the title of the manuscript needs to change because currently it emphasizes pandemic.

Authors' Response: Thank you- we have clarified.  All but one participant were still caregiving- one participant met the inclusion criteria (caregiver at least six months and within the last two years)- his care recipient passed away just prior to the start of the COVID-19 pandemic.   We changed the name of the manuscript to reflect that all participants were not caregiving during the pandemic- 12 of 13 were.  At the inception of this study there was no pandemic!   By the time we had IRB approval and were ready to start data collection, the pandemic was well under way and we felt it was essential to add a research question asking how caregiving changed during the pandemic.    Findings did not differ for the individual who did not care give during  the pandemic- he just did not answer interview questions pertaining to caregiving during the pandemic. 

Reviewer comment: Lines 111-112: Table 2 does not show self-reported health ratings, is this information accidentally omitted from the Table?

Authors' Response: Thank you- we've added this to the table.

Reviewer Comment: Section Findings: It seems that for the sake of coherency, the first paragraph should be divided in two paragraphs, the second one should start from "Data analysis revealed..." and this text should follow the Table 2 placement, not go ahead of it.

Authors' Response: Thank you- we made this needed change.

Reviewer Comment: •   Table 2: Correct the title, participants are in fact caregivers. In addition, cells for care recipients in the first column need to be revised. Further on - methods section states that demographics were collected, were age and the years of caregiving  the only two demographics you collected? Was there no information collected on socio-demographic characteristics? Education, employment, marital status, income etc.? If so, this needs to be noted as a limitation of this study. Also, you stated that you collected self-reported health rating - show it in this table.

Authors' Comment: Thank you- we made all needed changes to Table 2.

Reviewer Comment: Discussion: Please check whether the style of citing references in the text is in line with Journal's instructions.

Authors' Comment: Thank you- we have made our references AMA style.

Reviewer Comment: Lines 420-426: In this paragraph, consider underlining the possible impact on care recipients too, as a vulnerable population.

Authors' Response: Thank you- we added this important statement.

Reviewer Comment: Lines 434-437: Were information collected on this? These need to be presented in the Findings section, possibly within Table 2.

Authors' Response: Yes we collected income data and have added this to Table 2.

Reviewer Comment: Lines 445-447: Define the study criteria for inclusion of men within the Methods section.

Authors' Response: Thank you for this request- we have added study inclusion criteria.

Reviewer Comment: Lines 448-449: How was the effect of pandemic assessed for the one caregiver who did not have a caregiving role during the pandemic, as you've stated in the manuscript?

Authors' Response: Thank you- we have clarified that pandemic information was from only 12 of the 13 participants.

Reviewer Comment: English language of the manuscript: Please check the text for minor grammatical errors. E.g. in the last sentence of the Manuscript (in Conclusion)

Authors' Response: Thank you - we have spell checked more carefully!

Reviewer 4 Report

This article uses a qualitative descriptive study to describe the experiences of African American male caregivers during the COVID-19 pandemic.  Authors assert a need for this study due to the fact that the experiences of African American caregivers in the U.S. is lesser known, despite an increasing number of caregivers due to the aging of the population and a growth in aging research.

Feedback is as follows:

  1. Since the aim of the study is to “ explore and understand the experiences and perspectives of African American male caregivers, before and during a global pandemic” (line 36), the authors should consider stating ‘African American Male Caregivers’ in the title of the paper to reflect this.
  2. Line 43 – Since the statement “"in everyday terms of those events" is in quotes, perhaps a page number would need to accompany this citation.
  3. For participant recruitment (lines 49-55), what was the response rate?
  4. The authors have a good mention of participants receiving an incentive for participation and of participants selecting the interview dates and times to confirm to caregivers’ schedules.
  5. For Data Collection (lines 56-60 and lines 79-83), the information is repeated twice.
  6. Line 116 – For Table 2 - Participant and Caregiver Characteristics, where there any other sociodemographic data collected? For example, level of education, income, occupation, marital status, and related variables could provide further insight about the study sample.  Authors mention later in the paper “The sample represented demographic findings similar to other studies regarding education, health status, and employment” (lines 435-437).  If available, these variables should also be reflected in Table 2.
  7. The paper should be organized so that the themes and subthemes are distinguished from each other. For example, in lines 120-122, the subthemes to the theme ‘My time to take the reins’ should be presented as subheadings so that the themes and subthemes are distinguished from each other.  This should be the case for other themes in the paper (for example, ‘It’s a Male Thang’ in line 161). 
  8. Lines 128-133 – This is a profound and moving quote.
  9. Authors can also consider presenting some of the quotes within a table to organize and illuminate the information. Otherwise, some compelling quotes (e.g., lines 206-211) may be lost within the text.
  10. Line 253-310 – There is insightful information presented within the ‘Caring Amid a Pandemic’ section.
  11. Line 343 – With the statement “personal benefits to caregivers’ health outcomes”, are there caregiver quotes to provide further insight into this finding?
  12. Line 424- This is a compelling finding that among participants “nearly 40% opposed accepting the immunization”. Where there probes to gain more insight into this occurrence?
  13. Line 430 – Since “many of the caregivers reported symptoms of depression”, were participants referred to appropriate services?
  14. For the Conclusions, authors should expand on this section with future directions for research as well as potential policy implications of the research.
  15. Line 493 – Check for extra space in the Reference list.

Overall, this is an insightful, pertinent, and unique study on a very important topic.  It is pleasing to see a paper on this topic and the authors should be commended on their work.  There are areas of the paper including the Methods, Results presentation, and Conclusions that need to be expanded.  Attending to the clarifying questions may help to improve the overall paper.

Author Response

Reviewer Comment: Since the aim of the study is to “ explore and understand the experiences and perspectives of African American male caregivers, before and during a global pandemic” (line 36), the authors should consider stating ‘African American Male Caregivers’ in the title of the paper to reflect this.

Authors' Response: Thank you- we have changed the study title to be more accurate.

Reviewer Comment:  Line 43 – Since the statement “"in everyday terms of those events" is in quotes, perhaps a page number would need to accompany this citation.

Authors' Response: Thank you- we added the specific page number.

Reviewer Comment:   For participant recruitment (lines 49-55), what was the response rate?

Authors' Response: There is no response rate since we did not directly approach any participants and used flyers and social media to recruit.

Reviewer Comment: The authors have a good mention of participants receiving an incentive for participation and of participants selecting the interview dates and times to confirm to caregivers’ schedules.

Authors' Response: Thank you!

Reviewer Comment: For Data Collection (lines 56-60 and lines 79-83), the information is repeated twice.

Authors' Response: Thank you for pointing this out- our error!

Reviewer Comment:  Line 116 – For Table 2 - Participant and Caregiver Characteristics, where there any other sociodemographic data collected? For example, level of education, income, occupation, marital status, and related variables could provide further insight about the study sample.  Authors mention later in the paper “The sample represented demographic findings similar to other studies regarding education, health status, and employment” (lines 435-437).  If available, these variables should also be reflected in Table 2.

Authors' Response: Thank you. We did collect this data and have added it to Table 2.

Reviewer Comment:  The paper should be organized so that the themes and subthemes are distinguished from each other. For example, in lines 120-122, the subthemes to the theme ‘My time to take the reins’ should be presented as subheadings so that the themes and subthemes are distinguished from each other.  This should be the case for other themes in the paper (for example, ‘It’s a Male Thang’ in line 161). 

Authors' Response: Thank you- we added subheadings.

Reviewer Comment:  Lines 128-133 – This is a profound and moving quote.

Authors' Comment: Yes we agree!

Reviewer Comment:   Authors can also consider presenting some of the quotes within a table to organize and illuminate the information. Otherwise, some compelling quotes (e.g., lines 206-211) may be lost within the text.

Authors' Response: Thank you for this great idea- we added the additional table.

Reviewer Comment:  Line 253-310 – There is insightful information presented within the ‘Caring Amid a Pandemic’ section.

Authors' Response: Thank you

Reviewer Comment:  Line 343 – With the statement “personal benefits to caregivers’ health outcomes”, are there caregiver quotes to provide further insight into this finding?

Authors' Comments: Thank you- we added additional quotes.

Reviewer Comment:  Line 424- This is a compelling finding that among participants “nearly 40% opposed accepting the immunization”. Where there probes to gain more insight into this occurrence?

Authors' Response: Unfortunately we did not probe more- at the time of data collection there was great vaccine hesitancy- especially among African Americans in the U.S. The data collector is African American and did not want to "push" hesitancy.

Reviewer Comment:   Line 430 – Since “many of the caregivers reported symptoms of depression”, were participants referred to appropriate services?

Authors' Response: This is a very important point. The first author, who was the data collector, is a very experienced registered nurse who assured all participants were receiving appropriate healthcare. All participants were given the number to a crisis hotline as well.

Reviewer Comment:  For the Conclusions, authors should expand on this section with future directions for research as well as potential policy implications of the research.

Authors' Response: Great suggestion- we have added policy implications.

Reviewer Comment:  Line 493 – Check for extra space in the Reference list.

Authors' Response: Thank you- we cleaned up the reference list.

Reviewer Comment:  Overall, this is an insightful, pertinent, and unique study on a very important topic.  It is pleasing to see a paper on this topic and the authors should be commended on their work.  There are areas of the paper including the Methods, Results presentation, and Conclusions that need to be expanded.  Attending to the clarifying questions may help to improve the overall paper.

Authors' Response: We are grateful for your careful and thoughtful remarks to enhance this manuscript. 

Round 2

Reviewer 1 Report

well done. All suggestions had been done

Author Response

Thank you for your second review. We are appreciative. 

Reviewer 2 Report

The paper is now okay.

Author Response

Thank you for your second review. We are appreciative!

Reviewer 3 Report

I would like to thank the Authors for revising their manuscript.

  Abstract: Rephrase "COVID-19 virus" - rather say "in relation to vaccination against COVID-19". Methods, Participants: Was there a specific geographical location (e.g. town, state) from which participants were considered for recruitment? Line 157 mentions a region, was this your target region or? Methods: Since on Lines 111-112 you state that interviews lasted 49-99 minutes, please align information given throughout the Methods - namely text on line 77. Methods, Lines 91-92: What does sending study materials to a "select group" mean? Clarify selection. Regarding Authors' response that "At the inception of this study there was no pandemic!" - this response is not clear to me. First version of the manuscript did not give many details on time frame of study, and study emphasized pandemic right from the title - so it was not possible to deduct that there was no pandemic when study started - plus 2021 was given as year of data collection. Now we can see it was conducted over the February-April 2021 period. Pandemic started at the beginning of 2020. What do Authors refer to as "inception of study" in their response? Do they refer to planning the study and obtaining IRB Approval - when was that? In addition, you stated in your same reply that you added questions after obtaining IRB approval - did you obtain an amendment approval for that? Please clarify these issues which were raised by your response.

Author Response

 Reviewer Comment: Abstract: Rephrase "COVID-19 virus" - rather say "in relation to vaccination against COVID-19".

Authors' Response: Thank you- we have made this change.

Reviewer Comment: Methods, Participants: Was there a specific geographical location (e.g. town, state) from which participants were considered for recruitment? Line 157 mentions a region, was this your target region or?

Authors' Response: Thank you- we have provided more specific detail on the region.

Reviewer Comments: Methods: Since on Lines 111-112 you state that interviews lasted 49-99 minutes, please align information given throughout the Methods - namely text on line 77.

Authors' Response: Thank you. We have aligned the interview times.

Review Comments: Methods, Lines 91-92: What does sending study materials to a "select group" mean? Clarify selection.

Authors' Response: Thank you- we have clarified "select group".

Reviewer Comments: Regarding Authors' response that "At the inception of this study there was no pandemic!" - this response is not clear to me. First version of the manuscript did not give many details on time frame of study, and study emphasized pandemic right from the title - so it was not possible to deduct that there was no pandemic when study started - plus 2021 was given as year of data collection. Now we can see it was conducted over the February-April 2021 period. Pandemic started at the beginning of 2020. What do Authors refer to as "inception of study" in their response? Do they refer to planning the study and obtaining IRB Approval - when was that? In addition, you stated in your same reply that you added questions after obtaining IRB approval - did you obtain an amendment approval for that? Please clarify these issues which were raised by your response.

Authors' Response: We greatly apologize for this confusing language in our response.  We wrote our response in a conversational tone- This study was the first author's PhD dissertation study, thus the comment about "first inception there was no pandemic" referred to the author's PhD studies journey.   The first author had done a pilot study (pre-pandemic) with the interview guide in early 2020 and amended the IRB application (IRB approval obtained) to include the new questions about the Pandemic for this study. All 13 participants in this study had the same IRB approved interview guide.   The one participant who did not answer the pandemic question was included in the study because he still met all other study inclusion criteria - he had provided caregiving for at least 6 months within the last two years.   We  have clarified this in the manuscript. 

Reviewer 4 Report

The authors have done well to address the reviewer feedback.  The revised paper is clearer and more comprehensive, including the added information in the Introduction about African American male caregivers and information in the Discussion about policy considerations. 

One comment:

  1. Lines 38-39 – In “Although the non- referred literature is abundant…”, do authors means to say ‘non-refereed’?

The authors are to be commended for their efforts.

Author Response

Thank you for your great attention to detail with our typo- we have fixed it. We appreciate your reviews.